# An Investigation of Lecturers' Teaching through English Medium of Instruction—A Case of Higher Education in China

**Haijiao Chen [1], Jinghe Han [2,*] and David Wright [2]** 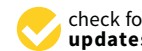

[1]   Center for Faculty Development, Huaqiao University, Xiamen 361021, China; star0868@hqu.edu.cn
[2]   Centre for Educational Research, Western Sydney University, Penrith, NSW 2751, Australia;
     david.wright@westernsydney.edu.au
*   Correspondence: j.han@westernsydney.edu.au; Tel.: +61-2-4736-0216

**Abstract:** Teaching through English Medium of Instruction (EMI) is a theory-based pedagogy that has been adopted in many European and Asian countries as a strategic initiative in educational internationalization. To date, there has been little research into EMI in-class teaching and learning. In effect, lived experiences in EMI in-class practice have been largely ignored. To address this gap, we reported on a case study that explored the linguistic and pedagogical characteristics of EMI lecturers' teaching in a university in southern China. Twenty academic staff in the university's EMI programs were recruited. Their in-class EMI teaching processes were observed and audio-recorded. The data was analyzed by drawing upon multilingualism and instructional design theories. This research found that Chinese EMI lecturers' bilingual repertoire led to their English instruction featuring Chinese language influences, from pronunciation to syntax and that translanguaging strategies were purposely employed to achieve their goals including students' cognitive understanding, affiliative bonds and the lecturers' own survival for teaching. Further, that and the instruction applied in the EMI classes were more topic-centered than problem-centered, focusing on activating new learning and knowledge presentation through demonstration.

**Keywords:** English Medium of Instruction (EMI); EMI teaching; instruction theory; multilingualism; L1/L2 transfer; translanguaging; first principles of instruction

## 1. Introduction

Teaching through English Medium of Instruction (EMI) is a fairly new learning delivery system. It has arisen as part of an emergent dynamic: the globalization of education within a global economy. It is described, and commonly accepted, as the "use of the English language to teach academic subjects in countries or jurisdictions where the first language (L1) of the majority of the population is not English" [1] (p. 2). Thus, it is not a major consideration in countries where English is the national language. Rather, it arises as the "business" of the rest of the world when governments and the education systems that they manage pressure teaching staff to use English as the medium of instruction with the purpose of raising their domestic students' English proficiency and/or making their class accessible to international student groups.

Teaching via EMI across Europe dates back to the 1990s and is evidenced in the Bologna Declaration [2] as an objective in EU tertiary education reform. EMI was seen as a strategic move for EU countries to internationalize their universities' curriculum in pursuit of accreditation in the fast-developing, globalized world [3,4]. With the increasingly competitive marketing of higher education in recent years, some major Asian countries and/or regions such as China, Vietnam, Korea

and Taiwan have been swiftly moving toward EMI delivery in their higher education sector for various but comparable reasons [5–7]. For example, the Chinese Ministry of Education (MOE) has recently introduced policies that prioritize EMI teaching in higher education as part of the country's strategic plan for developing its World *First Class University* and *First Class* Academic Discipline Construction [8,9]. This is an element of the nation's ambition to make China the top destination for international students [6]. Similarly, teaching through EMI has become a national movement [10] in Taiwan's higher education system, with the purpose of addressing the sector's declining enrolment rate through attracting international students [11]. For South Korea and Vietnam, their increasing number of international enterprises and growing demands for a skilled labor force with English proficiency requires employers, and therefore the education sector, to improve English language skills. The perception that English is essential to each nation's participation in a rapidly growing global economy is reflected in the rationales developed for promoting EMI teaching in their higher education systems [12–16].

This market-driven top-down initiative undertaken by numerous countries has not necessarily been an easy or smooth transition. The absence of clear understanding of the reasons for this is largely due to the lack of bottom-up data that looks into and informs educators, policy-makers and others about how the process of EMI teaching is actually enacted [1]. Insight of this kind is addressed here by drawing upon current literature and investigating EMI lecturers' teaching, particularly the features they demonstrate in practice. By arguing for this, our presupposition of EMI teaching is twofold. Firstly, as EMI teaching is carried out by lecturers through their secondary language (L2; in this case, English) to a student audience learning both a subject and a second language , the process is a more complicated matter than direct L1 to L2 translation [17]. In this respect it is important that lecturers' L1 and L2 relationship is scrutinized when investigating the features of their language use [18]. Secondly, and more fundamentally, what matters in EMI teaching is the choices lecturers' make in their pedagogical practice. Specifically, we argue that how lecturers design and implement their classroom instructions is the key to promoting learners' responses and making learning arise [19]. Thus, it is critical that EMI lecturers' classroom practices, through their instructions to students, is examined from a pedagogical perspective. A supplementary argument can be made that particular cultural and educational contexts needs to be considered in the process: necessarily, local practice is influenced, but not ultimately determined, by global presuppositions. It is accepted that the construct of EMI lecturers' instruction is "characterized on the basis of [their] philosophical beliefs" [19] (p. 27). These beliefs are influenced by their culture, their prior educational experience, and the assumptions of the educational system they are employed within [20–23]. The following section provides a review of literature that has informed this research.

## 2. Issues in EMI Teaching

Literature on EMI teaching tends to be concentrated into three major clusters: English language, pedagogy or teaching strategies, and peripheral studies such as perceptions and attitudes. To date, language proficiency has been the most heavily researched area in EMI studies. This is understandable, as much tension arises when an academic subject is delivered through a lecturer's secondary language. Current research in this area is primarily conducted through assumptions that suggest monolingual (in contrast to a multilingual) priorities. EMI pedagogy research is anchored in language schemes such as classroom discourse or the everyday use of language in the classroom. The practice of "instruction" through EMI has not been researched in a systematic manner. The third cluster is "about" EMI. It includes EMI lecturers' perceptions and beliefs about their role in EMI teaching [1,24] lecturers and students' attitudes towards EMI programs [25,26]; universities' opinion on the usefulness of EMI training [27]; national, institutional and personal thoughts about EMI [5]; and students' expectation of learning through EMI classes [28]. We call this "peripheral EMI research" as it does not address EMI teaching itself. Thus, in the following review sections, we focus on reviewing the two areas directly related to the core subject matter, "language" and "instruction" in EMI teaching.

*2.1. Language Issues in EMI Teaching*

Through reviewing recent research, one finding is that EMI lecturers' English proficiency is singled out as the key issue for EMI teaching [29–32] and this has caused growing concerns among institutions [1,3,4,16,33]. A research study conducted by a British research team across 55 countries worldwide found that most of the universities under investigation took English as the main, and often even the only, criterion when recruiting EMI lecturers [1]. Interestingly, most of these countries did not have a standardized English benchmark test, and assigned those who were believed to have good oral English to teach EMI programs [1]. A more recent study from Denmark has reported that, for quality control purposes, some universities have begun generating and implementing policies for the internal assessment of EMI lecturers' English proficiency [34]. For example, an oral performance test called the Test of Oral English Proficiency for Academic Staff (TOEPAS) was developed in a Copenhagen university to raise lecturers' awareness of the level of their English skills, aiming to address demands for quality assurance in EMI courses [34].

Existing EMI training programs also tend to highlight lecturers' English skills. Interestingly, training programs were often conducted by language experts from language centers instead of education faculties [4,35–38]. For instance, recent research conducted in Spain, explored eight universities' EMI training programs. It found most of the offers of training for EMI lecturers were English courses by language departments in "general English proficiency" and "academic English", or "Training Program for English in Teaching" [38]. This research also found pedagogical support in the form of providing grants or funding, but no specific training programs were in place. This left EMI lecturers to develop their own pedagogical practice. Remarkably, when seeking professional development, EMI lecturers tended to narrow their stress on language proficiency as well. A number of studies have reported that the majority of lecturers and teachers in the field are not even sure what other aspects besides language they should be looking at in their EMI teaching. Their specific concerns are around their own and/or students' non-colloquial and 'accented' English; lecturers' poor English communication with students; and students' misunderstanding of, and confusion around subject vocabularies in English [39–45]. In response to the lack of adequate training, some EMI lecturers believe that attending international conferences or short study tours to an English-speaking country are the solution to problems in EMI teaching [46–49].

English as the center of EMI research is sensible, as language proficiency is identifiable marker in academic provision. However, there is an obvious gap here. That is, all the research leans on the English product. Noticeably, EMI lecturers' bilingual repertoire is largely ignored in the current research on EMI teaching. There is an absence of valid data around EMI lecturers' language characteristics, based on a useful breakdown of identifiable language use in the classroom. Such ignorance may be due to the "tricky" labelling of "English" Medium of Instruction, or it could be an intentional choice, designed to assert a monolingual set of priorities. As bilinguals, EMI lecturers are not insulated from their students or each other [50]. On the contrary, their EMI teaching utilizes two languages, and their first language is unavoidably intertwined with, or bleeds into their English use, despite the latter being regarded as the "formal" language of instruction. It is expected that at many levels, L1 and L2 transfer and translanguaging processes can be seen to play a very significant role in the EMI classroom.

*2.2. Pedagogical Issues in EMI Teaching*

Beyond language, there exists a volume of voices on EMI pedagogy or teaching strategies. Researchers have called for quality pedagogical training as a solution to the problem of improving EMI lecturers' teaching skills for two decades. They persist in developing thoughtfully designed workshops or structured short courses [16,51–56]. However, there is no research demonstrating that successful pedagogical programs have been established and implemented in EMI lecturers' professional training [27,35,55]. In a recent European overview, there was little evidence signifying that EMI lecturers undertake training in methods of classroom practice. Rather, they were encouraged to improve their

own teaching: in this regard it becomes a personal issue rather than a critical or systemic one [56]. A recent international survey, conducted through 79 universities found that the pedagogy for EMI teaching was "far from being treated as an important issue" and there was not "sufficient attention to the training and accreditation of the teachers engaged in EMI" [27] (p. 557). Most universities offering a significant number of subjects through EMI, admitted that they did not provide training for their EMI lecturers [16,27,57]. Nevertheless, a couple of studies among these do report on methodological programs. One research team reported on a Spanish university providing an EMI taster course with a purpose of up-skilling lecturers' strategies in teaching. That was described as an attempt, but it was clear that the course was not designed to question, and perhaps amend, EMI teaching methodologies [16]. By comparison, an EMI program for a group of lecturers in Italy was reported as a successful pedagogical package. Key to the program was that participant lecturers were asked to present their teaching. They then received peers' evaluation and feedback. This program was regarded as "successful" because the focus of the program was for lecturers to reflect on their teaching practice, particularly on how they responded to students' needs and how they optimized their interaction with students [16].

Parallel with these, there has been some research into pedagogy conducted through Content and Language Integrated Learning processes (CLIL). There are subtle differences between CLIL and EMI, as the literature suggests. EMI is mostly used in the context of universities, whereas CLIL is used at all levels of education and the referred "language" is not necessarily English. Despite these differences, these two are placed under the same research umbrella by some scholars [38,58,59]. CLIL researchers are more aware of the "dual focus on language and content learning" [38]. Some researchers assert that generalized methodological training for CLIL lecturers is not sufficient; they propose a "bilingual or CLIL methodology" that can increase lecturers' awareness of how language may affect the construction of disciplinary understanding [38,58,60]. Others suggest that CLIL lecturers need to carefully consider appropriate scaffolds to make sure learners are not cognitively overloaded by content and language at the same time. Further, when providing scaffolding, it is argued that lecturers should focus on content and tolerate deviation of the language from its standard usage [59,61]. These proposals are a step closer to the development of a useful "CLIL pedagogy", as students' needs, their cognitive load and teachers' support role (e.g., scaffolding) are seen as the most important issues to be addressed in the development of such pedagogy.

The review of the studies above suggests that mainstream research into EMI (or CLIL) pedagogy is at the stage of rich discussion but is insufficient in the provision of proven and effective classroom strategies. However, little is known about valid designs and approaches developed from a characteristic-based analysis of EMI (or CLIL) lecturers' lessons. Nor is much in written about the kind of evidence-informed pedagogical training that should be provided to EMI lecturers [60]. This triggers this study to address the absence of data that can explicate the representative experience of EMI lecturers' in-class teaching with the intention of informing the development and design of appropriate EMI programs. It asks the following question: what linguistic and pedagogical features can be observed in the process of Chinese lecturers' teaching through English Medium of Instruction? The following section provides the analytical framework that underpins this research.

## 3. Multilingual EMI Teaching Framework

The "English" in EMI, from a multilingual perspective, is not a monolingual issue. Anglophone English may be used as a point of reference but it is not a norm to follow. EMI lecturers, like many other bilinguals, do not necessarily focus on standard English in their teaching. They work through their own localized varieties of English [62]. Expecting EMI lecturers to treat a native norm as the goal is neither desirable nor realistically achievable [63]. Multilinguals and bilinguals do not live in separate language capsules. Instead, they move between their two (or more) languages. They are constantly drawn to work with 'perceived' and 'assumed' cross-linguistic similarities [64] when processing information. L1/L2 transfer is a natural part of this [64]. Based on people's diverse and unequal experiences, transfer

can occur at various levels. Multi/bilingual speakers may sometimes borrow a word with a meaning, or part of the sound system, from L1, and at other times they may borrow sentence structures from L1. Transfer can occur in a range of phonological, morphological, syntactical and semantic forms [65].

Multilingualism sees the concept of "translanguaging" entailing border thinking and knowledge that is conceived from a multilingual or bilingual position [65]. When lecturers translanguage, they switch the language mode as required for effective pedagogical practice. This "goes beyond what has been termed code-switching although it includes it" [65] (p. 45), because in translanguaging the code-switching practice and a hybridity of languages do not happen randomly, but systematically and strategically. Besides, it incorporates other kinds of bilingual language use such as translation [65]. More importantly, the process of going back and forth between languages can come with a strong education purpose [66]. On one hand, translanguing functions as scaffolding to facilitate students' content learning; on the other hand, it models practice to students, suggesting how they can employ their bilingual resources in learning [67].

English Medium of Instruction (EMI), as stated in the introduction, is widely accepted "to teach academic subjects" through the "use of the English language": an additional language for both the teacher and the learners [1]. The "instruction" in EMI, referring to Dearden's definition, is "to teach academic subjects". However, neither Dearden nor other EMI researchers have specified what "to teach" means. Is it a method, a process, a procedure or is it an orientation? In learning psychology, researchers define instruction as "the process of deliberately manipulating the environment of an individual so that his (sic) [learning] behaviour is changed in a specified way" [19] (p. 28). This occurs in teacher–learner interactions around educational materials [68]. The design of instruction should aim for learning and change through development in knowledge, abilities, perceptions and skills [66]. According to Tennyson and Merrill, to make learning effective instruction should be presented following a simple to complex sequence or order of learning in order to mobilize learners' behaviours in four domains: emotional, psychomotor, memorization and complex cognitive [19].

Merrill has identified the underlying principles that are widely accepted as essential to theorists of instruction design. These are described as the "First Principles of Instruction". By "principle", Merrill refers to a universal law of a relationship. He argues that a relationship is "always true under appropriate conditions regardless of program or practice" [69] (p. 43). Merrill offers five instruction principles that promote learning. These are when:

- "learners are engaged in solving real-world problems" (problem-solving);
- "existing knowledge is activated as a foundation for new knowledge" (using learner's prior knowledge);
- "new knowledge is demonstrated to the learner" (teacher applying knowledge in practice);
- "new knowledge is applied by the learner" (learner receiving the opportunity to apply knowledge in practice);
- "new knowledge is integrated into the learner's world" (using learned knowledge in the real world) [69] (pp. 44–45).

It is argued that when these principles are implemented in instruction design learning will be promoted (effectively and efficiently). These can be implemented in any delivery system as they are related to "creating learning environments and products rather than describing how learners acquire knowledge and skill from these environments or products" [69] (p. 44). This means that these principles can be used to design instruction for EMI and CLIL programs or programs delivered in someone's first language, and these principles can be used as a framework to exam an EMI lecturer's instruction. The next section explains the methodology of this study.

## 4. The Study

This research project comprised a qualitative case study focusing on EMI lecturers in one university in a city in southern China. The university was selected due to its EMI program. Currently, about 100 academic staff are registered in the program. They include professors, associate professors and

lecturers in the fields of Biochemistry, Global Studies, Engineering, Physics, Mathematics, Medical Science, Marketing, Computer Science and Metaphysics. Data was collected through researchers' participant-observation, as informed by the literature. In this area of inquiry researchers are usually more removed from the study. They report on related but tangential EMI issues, such as participants' self-reporting [22,70,71]. "There is a dearth of research, using objective tests rather than self-report" [38] (p. 64). In this project data was gathered more directly. A total of 20 lessons from 20 EMI lecturers were observed and audio-recorded. All these lessons used a lecture–tutorial model of integrated teaching, which is the dominant delivery mode at the investigated university. This may be different from the separated lecture and tutorial mode used in many Western countries. Most of the observed classes contained 20–60 students. Observation enabled the researchers to capture the actual classroom situations (e.g., their language use and teaching strategies) and to document and analyze the EMI lecturers' ongoing teaching [46].

Two approaches to qualitative content analysis were employed in the data process: a Directed Approach and an Inductive Approach [72]. The first phase was through the Directed Approach. The analytical constructs were informed by the multilingual EMI framework, and the data were reviewed and coded against the pre-set concepts and categories. Coding at this phase was preliminary, which means coding was achieved immediately from raw data against these concepts and categories. The second phase was the Inductive Approach. Particular attention was paid to the data that was not foreshadowed by the theoretical framework. Data coding and categorizing were completed in an open manner at this phase. Instances of the emergence of new patterns and new questions were pursued.

## 5. Results

### *5.1. Findings in Language Use*

Data gathered in this research revealed the common occurrence of phonological, semantic and syntactic transfers from the lecturers' L1 to their English language instruction. This appears to shape principal characteristics of their EMI teaching. It became apparent that lecturers' English language use was strongly shadowed by the grammatical rules and semantic units of their L1. The conceptual patterns and linguistic codes characteristic of Chinese provided essential supports in their English language based instruction. Further, almost all the EMI lecturers created a translanguaging space to present a coordinated teaching performance, and to aid students to comprehend and construct desired learning. Translanguaging was, in effect, the pedagogical choice of most of the lecturers' in their EMI classes.

### 5.1.1. Grammatical and Semantic Transfer

Across the two languages (Chinese and English), the EMI lecturers consistently perceived "fake" similarities in the areas of pronunciation, phonology, word meanings and grammatical rules. Tables 1 and 2 provide some examples.

**Table 1.** Transfer of grammatical and semantic meaning.

| Group | Excerpts (Sentences in Brackets Are Adjusted for Correct English Usage) | Codes |
|---|---|---|
| Group 1 | "What it look like?" (What does it look like?)<br>"Peers can help?" (Can peers help?)<br>"So, anybody tell me?" (Can anyone tell me?)<br>"I ask you again." (I will come back and ask you about this later.)<br>"Follow me! Follow me?" (Do you follow me?) | Question sentences |
| Group 2 | "If do this … it connect xxx. How to connect the first and second?" (If we do this, it will connect to xxx. How can the first and the second then be connected?)<br>"Without US the police what would the world be? Peace? Stable? Would be in chaos." (If we didn't have the US policing the world, what would the world be like? Would it be peaceful? Stable? It would be chaos.)<br>"We need to think about why different view for governing the country" (We need to think about why there are different views for governing a country.) | Subject-less sentence |

**Table 1.** *Cont.*

| Group | Excerpts (Sentences in Brackets Are Adjusted for Correct English Usage) | Codes |
|---|---|---|
| Group 3 | "Which one is special characters?" (Which ones are special characters?) | Singular/plural form |
| Group 4 | (The lecturer wrote "patriotism" on the whiteboard) "Do you know what is it?" (Does anyone know what this means?) | Sub-clause structure |
| Group 5 | "Now open your computer." (Now turn on your computer.) "Good! You've got sharp eye! ("Good! You see things clearly.") "Just speak out. I don't like class quiet". (Speak up! I don't like a quiet classroom.) | Semantic transfer |

The above examples reflect the reliance on the sentence structures of the first language of the speakers, which created errors in language transfer. Consistently incorrect structures that were observed included:

- The structure of question sentences: Sentence structures were often seen following the Chinese pattern of adding a rising tone and question mark at the end of an assertive sentence (see example in Group 1).
- Statements without a subject: This occured as there are no strict subject-predicate rules in Chinese sentences and a sentence can be valid without a subject (see Group 2).
- Lack of clarity with reference to singular and plural: There is no consistency in singular and plural use in Chinese subject and predicate relationships, and some of the lecturers transferred this usage into their English expressions.
- Lack of clarity in sub-clause structures: Overuse of conjunctions reflected the influence of particular Chinese language expressions (see Group 3).
- Semantic transfer: This occured when Chinese concepts or terminologies do not have English equivalents. For example, "turn on" was translated into Chinese "open", thus "turn on the computer" becomes "open the computer", and "turn on the light" becomes "open the light".

5.1.2. Phonological Transfer: Consonants, Vowels and Consonant–Vowel Complex

The data demonstrated phonological transfer in the pronunciation of particular English-language words in the lecturers' English Medium of Instruction. This was the case even with speakers who were fluent in English in the classroom. The main problem was with English consonants and vowels that are absent in Chinese (the speakers' first language), and those words that end with friction consonants. Examples from the observation data appear in Table 2 below.

**Table 2.** Phonological transfer: consonants, vowels and consonant–vowel complex.

| Consonants: /ð/ /θ//dʒ//tr//tʃ//l/ | | Vowels: /eɪ//e//aɪ//eə//ʊ//æ//ʌ/ | | Chinese Consonant-Vowel Combining /f//t//d//s/ | |
|---|---|---|---|---|---|
| Lecturers' pronunciation | English word | Lecturers' pronunciation | English word | Lecturers' pronunciation | English word |
| [zan] [zi:si] [ze] [aze] | then this the other | [du/u] [tu/u] | do too | [gai/si] [pla/si] [i/fi] [i/zi] | guess plus if is |
| [sing] [sink] [sru:] | thing think through | [Min] [dou/min] | main domain | [hai/de] [gu/de] [an/de] [in/ste/de] [nide] | had good and instead need |
| [chuans-] [Chuanpu] [machi] | transform Trump much | [uen] [en] | when in | [di/li/te] [ei/te] [ba/te] [krei/te] [dao/te] [gai/te] | delete eight but create dot get |
| [an/gou] | angle | [dang] | done | | |
| [jia/ste] | just | [wai/ri/bao] | variable | | |
| [you/ruo/li] | usually | [pai/er] | pair | | |
| | | [san] [kou/san] | sine cosine | | |

In these examples, it was explicit that the lecturers' L1 directly influenced their L2 pronunciation. The English consonants /ð/, /θ/, /dʒ/, /tr/, /tʃ/ and /l/ were replaced by the Chinese sounds [z], [s], [j], [ch], [q] and [o] (see column 1 and 2 in Table 3). This was due to the absence of these English sounds in the Chinese pronunciation system. The English vowels /eɪ/, /e/, /aɪ/, /eə/, /u/, /æ/ and /ʌ/ were changed to [i], [en], [an], [ai], [u], [ai] and [ang] in some lecturers' pronunciations (see columns 3 and 4 in Table 3). It was also observed that a significant proportion of the lecturers would tend to add the vowel sounds [e] and/or [i] to words ending with silent consonants /f/, /t/, /d/and /s/ (columns 5 and 6 in Table 2 above). This appeared to be due to some characteristics of the Chinese pronunciation system. The majority of the sound system of Chinese words are structured as a consonant–vowel complex. This means that Chinese words end with vowels instead of consonants. Some of the lecturers were observed adding a vowel at the end of those English words ending with a consonant, influenced by their familiarity with Chinese pronunciation. These findings indicate that the EMI lecturers applied their previous phonological knowledge in the use of new language. Some phonology researchers regard this as the result of L1 interference and label it as a negative transfer, arguing that this is the source of error [73]. Our argument is that such "negative" transfer plays a positive role in EMI teaching for two reasons. Firstly, although imperfect, it acts as a stepping-stone for lecturers to deliver the content; secondly, by sharing the same L1 between the lecturers and their students, such transfer may exist within the student group, and thus the lecturers' inaccurate pronunciation may not negatively impact their students' understanding.

**Table 3.** Translanguaging to scaffold learning.

| N. | Excerpts | Function |
|---|---|---|
| 1 | Teacher (T): We have mentioned this in our last class. Table of coding. 编码的表 | Partial translation (Emphasis) |
| 2 | T: If we just use two variables, can you think about it?<br>Student (S): (no response)<br>T: 想一想怎么用两个变量求值 | Meaning translation (Reiteration) |
| 3 | T: How do we determine the interval of convergence for a power series?<br>S: (silence)<br>T: 收敛区间。当时我们讲的是 . . . .谁能回忆一下? | Partial translation (Cluing) |
| 4 | T: I don't know whether you finished your homework. Have you done it?<br>S: (silence)<br>T: 感觉有困难吗?<br>S: 有。 | Code-switching (making emotional connection) |
| 5 | T: This is ehhh . . . 这是随意性，跟过程就没关系。 | Code-switching (Sense making) |

### 5.1.3. Translanguaging Strategies

The use of translanguaging strategies was evident in these lecturers' EMI class. They were used particularly for the purpose of highlighting key points for students' comprehension, lecturers' meaning-making through negotiation of the two languages and for making affiliative connections with students (Table 3). Three types of translanguaging were found: partial word-by-word translation, meaning translation and inter-sentential code-switching.

In Excerpts 1 and 3, the lecturers partially translated what was said in English into Chinese. In Excerpts 4 and 5, lecturers used inter-sentential code-switch, or switched code between sentences. Excerpt 2 is an example of meaning translation. These translanguaging strategies reflected lecturers' intent to scaffold students' learning. This included emphasizing a key point (Excerpt 1), reiterating the meaning (Excerpt 2), providing a clue (Excerpt 3), and making teacher-student connection by asking students' their feelings about the homework (Excerpt 4). Excerpt 5 reveals the lecturer switched to Chinese to negotiate meaning due to difficulties in finding an appropriate English expression. This can be described as a sense-making process. The data also demonstrates that, despite their lecturers' language code, the students' answers were almost always in Chinese. Even when the answer could be as simple as "yes", they would answer "有". Excerpt 4 is an example of this.

*5.2. Pedagogical Findings*

Using Merrill's First Instruction Principles to examine the data, some common pedagogical features were found in all lecturers' EMI teaching. Three pedagogical approaches were most frequently used: engaging students through questioning, presenting information accompanied with reasoning explanation and/or examples, and repeating information by translating between L1 and L2 as the main scaffolding strategies. A few aspects of Merrill's First Instruction Principles were absent, including problem-centered instruction, and the application and integration of new knowledge into practice.

5.2.1. Engaging Learning through Questioning

The data reveals that EMI lecturers tended to activate new knowledge learning through asking students to recall what had been learned. This was the main method of initiating lecturer–student interactions. It was observed that no opportunities were provided for students to work with each other. The lecturers initiated questions with three types of intention: fact-checking (Excerpt 1), classroom procedure (Excerpt 2) and activating new learning (Excerpts 3 and 4) (see Table 4).

**Table 4.** Seeking interaction through questioning.

| N. | Excerpts | Question Type |
|---|---|---|
| 1 | T: "Did you read the two articles I sent you?" <br> S: . . . (Silence). <br> T: "Did you? Did you?" <br> S: . . . (No answer). | Fact-checking |
| 2 | T: "Do you follow me?" <br> S: . . . (No answer) <br> T. "Do you follow me?" <br> S: . . . (No answer) <br> T: "So you cannot follow me? . . . Just speak out if you can't. It doesn't matter." <br> S: . . . (No answer). | Class procedure |
| 3 | T: We went through the superpowers' leadship model. Can I ask someone to give a brief on that? <br> S: . . . (Silence) <br> T: Can anyone say something on that? Anybody? <br> S: . . . (No answer) | Cognitive question (activate learned knowledge) |
| 4 | T: Can you think about two good things about Python programming? <br> S: . . . (Silence) <br> T: any idea on this before we move on? <br> S: . . . (No answer) | Cognitive question (activate learned knowledge) |

As the data suggests, some of the questions asked (Excerpts 3 and 4), required students to give an opinion or judgement based on their knowledge base. Such questions were asked to challenge students' ways of thinking. In other cases (Excerpts 1 and 2), the questions were either used for fact-checking or were strictly procedural, with a "yes" or "no" type of answer expected. As the data indicates, most of the questions received a silent response despite the various types of questions asked. Whether or not the students were engaged in the subject matter is hard to know due to their quietly passive behavior.

5.2.2. Reasoning the Explanation with Demonstration

It was observed that the majority of lecturers, when explaining concepts, formulae and procedures, did so through describing a cause-and-effect relationship, often accompanied with the provision of examples (see Tables 5 and 6). Instead of "remember what I tell you", the lecturers tried to explain the reason and their logic (Excerpts 1 and 2) (see Table 5), and show "how" and "why" with examples. This strategy enabled students to follow and map the structures of knowledge more easily.

**Table 5.** Reasoning the explanation with demonstration.

| N. | Excerpts | Codes |
|---|---|---|
| Excerpt 1: | " … that's why we should … " | Explaining the reason |
| Excerpt 2: | "So how can this happen? … Let's look here (pointing at a formula on the PPT). We have the formula (xxxx). If you start from (X) you will get (Y)." | Explaining the logic |
| Excerpt 3: | "How should we select … ?" … (Silence … ) "How?" … (No answer). "Let me give you an example … " | Showing how |
| Excerpt 4: | "This is very important! Let me explain it. When you … " | Showing why |

### 5.2.3. Repetition as the Key Scaffold

The use of repetition as a key strategy was observed frequently in the teaching repertoire of the lecturers. Repetition was found to serve three purposes: to encourage students to give their opinion (Excerpt 1), to draw students' attention (Excerpt 2) and to reinforce learning (Excerpt 3). This strategy was sometimes accompanied by direct translation (Excerpt 2).

As discussed earlier, minimal teacher–student interaction was observed in most of the EMI classes, and only very occasionally did students respond to lecturers' questions. This made repetition a particularly important strategy in teaching, as the lecturers received little response from students.

**Table 6.** Repetition strategy.

| N. | Excerpts | Codes |
|---|---|---|
| Excerpt 1 | T: Which one do you choose? Which one? Which? I like to hear your voice. | Repeating to encourage students to give their opinion |
| Excerpt 2 | T: How should we select xxx? How?<br>S: (No response)<br>T: This is very important!<br>怎么选？这个很重要！ | Repeating to draw students' attention |
| Excerpt 3 | T: What are the types of distribution we learned last week?<br>S: Uniform, normal and 卡方.<br>T: Yes. The uniform distribution, the normal and the Chi-square, and F distribution. | Repeating for reinforcing the learning |

## 6. Discussions

### 6.1. Chinese EMI Lecturers' Language Characteristics

Transfer theorist Ringbom argues that when the learners' L1 and L2 belong to the same language family (e.g., English and German), they tend to share more linguistic similarities; thus, transfer between two such languages is more likely to occur. For Ringbom, English and Chinese are distant languages in formation and development, and they are in "zero" relation. It is argued, therefore, that there is a minimal linguistic transfer in learning for Chinese–English bilinguals [64]. The results of this research do not support this proposition. Following Ringbom's transfer and language distance theory, Chinese EMI lecturers would have had little to draw upon from their Chinese linguistic repertorie. However, the data gathered reveals that transfer occurred at all levels from pronunciation to syntax. Thus, it can be argued that it is not how related or unrelated English and Chinese are that matters; it is how the bilinguals perceive the relationship that matters most to them in language transfer. The data further demonstrates a number of negative transfers. These occurred because of lecturers' inaccurate (or "fake") perception of language similarities. Further, these inaccurate similarities shaped the language features of the Chinese EMI lecturers. They included English pronunciations with an integration of Chinese versions of consonants and/or vowels, English words with a given Chinese meaning, and English expressions following Chinese language rules. This finding supports the theoretical standpoint made in the review section: "English Medium of Instruction" is not and will never be a monolingual

issue, despite the assumptions suggested by its name. Unfortunately, the current research reported in the literature section seems more interested in EMI lecturers' or teachers' perfect English, and is focused on "standard English" [1], training for academic English [38] and learning from native English speakers [46,48,49]. Our argument is that EMI lecturers are bilinguals, and when their English is examined their full language repertoire should be scrutinized from a multilingual perspective. Excessively requiring or addressing perfect English may put EMI lecturers at risk of giving priority to language and sacrificing the teaching content.

Beyond L1/L2 transfer, translanguaging is identified as explict pedagogical practice occurred in Second Language learning [18,64,66]. Garcia argues that moving between languages in a bilingual class is pragmatically essential [65]. Aligning with Garcia, Baker asserts that translanguaging involves making use of bilingual teachers' and students' linguistic and cognitive resources, to help learning achievement; thus, translanguaging is "cognitively, linguistically and operationally sensible" [74] (p. 229). Garcia further argues that bilinguals do not show clear-cut decisions around the hybrid use of the two languages [65], and a few other scholars also believe that translanguaging is a blurry zone for open negotiation [66,75]. Such studies might give the impression that translanguaging does not involve design behind it. This seems to suggest that translanguaging is a natural and random occurrence in bilingual class [76]. However, among the EMI lecturers studied in the present paper, their translanguaging use showed distinct patterns. As demonstrated in the data display section, three types of translanguaging were found: partial word-by-word translation, meaning translation and inter-sentential code-switching. There was consistency in this. Additionally, two types of translation were found in situations where the lecturers introduced new concepts or formulae. Translation was used to make sure students understood the content. Code-switching occurred when lecturers lacked English expressions and when the lecturers tried to build personal connections with students through asking them about their feelings. The use of translanguaging to connect students' sense of belonging for "affiliative" purpose was also reported as a practice in English as a Second Language (ESL) class [75] (p. 128). Therefore, it can be seen that the EMI lecturers observed in this research project used translanguaging to serve three aspects of the teacher–student relationship: students' emotions, students' cognitive understanding and the lecturers' survival strategies. They demonstrated that any movement between L1 and L2 was to facilitate effective content understanding. This differs from a pedagogical practice [70] in ESL that mostly sees language learning as the principal focus, along with an orientation towards perfect English.

### 6.2. Chinese EMI Lecturers' Pedagogical Features

As the data indicates, of Merrill's five instruction principles, two were observed in use by the EMI lecturers: activating the learning and explanation with demonstration. According to Merrill's second principle, "learning is promoted when relevant previous experience is activated" [69] (p. 45). The data demonstrated the lecturers' use of this principle in their teaching (Table 4). Before new knowledge was introduced, they tended to ask students to recall what they have learned or know in relation to the new topic. This was with an explicit purpose, that of assisting the students to lay the foundations for their new learning. Besides, in accordance with Merrill's third instructional principle, learning is effective when "new knowledge is demonstrated to the learner" [69] (p. 45). Enactment of this principle was observed in the EMI lecturers' teaching (Table 5) as well. The data reveals that the lecturers tended to begin by presenting general knowledge, theories or concepts (e.g., formulae or universal rules). The presentation was often followed by reasoning explanations and "showing how" to apply the knowledge in practice through clear procedures and examples. This practice aligns with Merrill's argument that "effective instruction is never presenting 'remember-what-you-were-told' information [69] (p. 45). Instead, "demonstrating how" is the key when instructing new knowledge learning.

Three other principles of Merrill's instruction framework were hardly found in the EMI lecturers' teaching. Merrill's fourth instructional principle emphasizes opportunities and guidance for learners "to use their new knowledge or skill to solve problems" [69] (p. 46), and the fifth principle is to provide

opportunities for students "to integrate [transfer] the new knowledge or skill into their everyday life". These two principles relate to the transference of learning responsibility from lecturers to students. Students were not guided through opportunities to reflect and explore the use of learned knowledge in practice, although lecturing–tutoring integration is the mode of the observed EMI classes, and, in which case, students should be the center of the transference. The first, and a more overarching principle of Merrill's, is "problem-centered instruction" [69] (p. 45). This emphasizes a holistic task through one lesson. According to this principle, learning objectives should be introduced to students at the beginning of a lesson and all the activities should be linked and directed to the completion of the task and the achievement of objectives. However, most of the EMI lecturers organized their class through topic-centered instruction. Teaching components were in isolation rather than related to a task or a problem set to be solved or completed. Unsurprisingly, due to the absence of the first principle, there was an absence of knowledge application and integration (the fourth and fifth principles) in the observed EMI classes.

As identified from the data, Chinese EMI lecturers' teaching is generally topic-centered. That is, lecturers started by introducing the teaching topic and finished the teaching after presenting new knowledge or information, often through demonstration. Lecturers did not empower students to actively apply and integrate knowledge and skills in practice. Such absence has determined students' passive position in learning. Merrill argues that whether a teacher prefers a problem-centered or topic-centered teaching design is largely determined by the educational philosophy that the teacher holds [69]. From the perspective of Hofstede's dimensional paradigm on culture and teaching [69], there seems to be a good reason why most of the EMI lecturers employed topic-centered teaching. By this paradigm, China is categorized as a society of High Power Distance (indicating a clear social hierarchy) where teachers are seen as the 'sage on stage'; thus, they dominate the talk in class. Comparatively, countries such as Australia, the UK and the USA are regarded as Low Power Distance societies [21] where teachers are 'the guide on the side' and they encourage active learning and expect students to be at the center [77,78]. The paradigm seems to give considerable insight into students' low response rate to lecturers' questions. English language might add additional barriers to class participation [23,52], but when the lecturers were center, and students were not genuinely given opportunities to participate, the class becomes lecturers' solo show. We argue that EMI lecturers' teaching is not a pedagogical issue that can reach an agreement across education systems of linguistic and pedagogical variety, it is a pedagogy that should be instructed by showing respect to effective learning and available local resources.

## 7. Conclusions

This research explored a group of Chinese lecturers' teaching through English Medium of Instruction. Distinctive features were found in three aspects: transfer, translanguaging and instruction. Firstly, these lecturers' bilingual repertoire enabled L1/L2 transfer to occur widely and made their English instruction feature Chinese language from pronunciation to syntax. Secondly, the EMI lecturers demonstrated three types of translanguaging strategy to achieve their prioritized functions in relation to: students' cognitive understanding, affiliative bonds and their survival in EMI teaching. Thirdly, beyond language, EMI lecturers demonstrated more topic-centered than problem-centered teaching pedagogy. They focused on activating learners through engaging their existing knowledge and presenting information to students through demonstration. There was little evidence of lecturers passing on the responsibility for students' knowledge application and integration.

This research suggests that two key insights deserve attention in future EMI training and research: EMI lecturers' pedagogical development can include the designing and practicing of problem-centered teaching; in terms of research, further studies of EMI lecturers' teaching can be conducted through comparing observation data from classes delivered by lecturers in English and their first language.

**Ethics Approval:** The study was conducted in accordance with the Declaration of Helsinki, and the protocol was approved by the Ethics Committee of Western Sydney University (Project identification code: H13012). All the original data (20 hours of audio recording) are saved at the Library of Western Sydney University.

**Author Contributions:** Three authors contributed equally. J.H. conducted the research design, H.C., J.H. and D.W. collectively worked on the project through data collection, analysis and writing. All authors have read and agreed to the published version of the manuscript.

**Funding:** This paper partly reports the findings of the project The development of EMI Pedagogy, a collaborative research between Western Sydney University and Huaqiao University funded by Huaqiao University (Grant number P00025138).

**Conflicts of Interest:** The authors declare no conflict of interest.

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
