# Peer review of "An Investigation of Lecturers’ Teaching through English Medium of Instruction—A Case of Higher Education in China"

_sustainability, doi:10.3390/su12104046_

Round 1

Reviewer 1 Report

This manuscript does not appear to be within the scope of the journal Sustainability, as I do not see any discussion within the paper addressing topics directly related to sustainability.

Also, regardless of the scope issue, it is not very clear what the novel contribution of the study is. The abstract presents the study's main contribution as showing that "EMI teaching is a complicated issue" -- but this was already self-evident. Beyond that, the abstract does not describe any specific results. The Conclusion describes some more, but many of these findings are not novel; for example, the observation that instructors' English is influenced by features of their first language is just an observation of language transfer, which is a phenomenon that has been known and well-described by linguists for over half a century (see, e.g., https://en.wikipedia.org/w/index.php?title=Language_transfer&oldid=936304181).

Author Response

thanks for the valuable comment.

Reviewer 2 Report

The paper discusses English medium instruction classes in higher education. The language transfer – grammatical, semantic, phonological and pragmatics is studied in communication of non-native English speakers – teachers and students. The influence of L1 in classroom interaction is discussed.

The chosen topic is relevant. The idea expressed in line 15 that little research has been done into EMI in-class teaching and learning is valuable. The paper outline is logical. The literature review and research theoretical background seem sufficient; the research methods are consistent with the research task (section 3 line 133): what features have EMI demonstrated in the process of teaching?

The analysis of grammatical, semantic, phonological aspects and pragmatics of L2 interaction is performed accordingly. However, the idea of specific communication settings – higher education classroom needs further detailing otherwise it can be concluded that in regular conversations the same language transfer may take place due to languages transfer mechanisms. More specifics or expectations related to educational contexts revealed in language transfer studied may be provided.

Section 3 lines 143- 144 says “The university has implemented EMI teaching in a range of subject areas for five years and currently about 100 academic staff are registered lecturers in the university’s EMI program”. It seems beneficial to add information about actual number of research participants – academics and students; format of classes where the discussed types of interaction took place. It can be considered if a minimum acceptable number of participants is applicable for such research.

It seems clarification is needed if the intention was to discover varieties of L1 influence or actually identify most common patterns of classroom interaction in which transfer reveals itself and their frequency. In this respect have other factors affected the study:  if students are mature learners – master degree and higher or bachelor degree and for how long have they been taking EMI classes – at what moment were they recorded; professional teaching experience of teachers (what if a teacher has not mastered strategies to engage students in interaction even in native language – there are only two reasons mentioned for students being silent in class in lines 442- 443 – cultural habit or English barriers; there may be others – e.g. not provocative nature of a task given to students that is pedagogical competences of a teacher), nature of discipline, learning formats – colloquium, seminar, and lecture – where students input may vary, etc.?

Phonological transfer is something what takes place outside classroom settings as well. It seems comments are needed why is has been considered. This transfer does not depend on classroom or any other communicative settings. It seems the results of phonological transfer presented in this paper need more clarification in relation to classroom settings interaction.

It is advisable to consider if one term should be used through the paper. Currently there are three: in Section 4.4 line 354 Pragmatic strategies; in 4.4.4.line 396 pragmatic educational strategies; line 355 and line 426 pedagogical strategies. Do they mean the same?

 It seems beneficial to comment what is understood by the pedagogical strategies – if they are related to classroom interaction or teaching methods and styles.

In Section 4.4. line 355 Data demonstrate  that around 30% of lecturers  fully developed pedagogical strategies in their English instruction. Please specify what data is meant. What is understood by fully developed pedagogical strategies? Is 30% of lecturers satisfactory number for this research or how should be it treated?

The sentence seems vague: line 359 It needs to be noted that the majority of the group demonstrated isolated use of these strategies. As follows from this sentence, there are some expectations ranging from key pedagogical strategies (line 426) to isolated use of pedagogical strategies. What are the expectations and how research data relates to them?

Some statements need clarification how such conclusion was reached, e.g. section 4.1 line 197 “there were very few in-class dynamic activities recorded. There were no opportunities for students to work with each other.” It seems beneficial to mention if it is related to specifics of teaching format, e.g. lecturing, to a teacher experience to organize classroom interaction as such? Perhaps it is worth commenting the teaching format of classes where recordings were made. E.g. lecture format might presuppose little direct interaction of students and a teacher via discussions and may be disregarded within this research due to this reason.

The paper abstract lines 21- 22 say This research is expected to provide insight for the development of localised institutional guidelines for EMI teaching and lecturers’ professional development in EMI teaching. There is no further mentioning if this objective has been achieved, partially or to full extent or needs further investigation.

In conclusion line 424 the implication is given that imperfect English does not stop EMI lecturers from making an effective teacher. This idea seems sound however clarification is needed what is understood by effective teacher in the context of this research. How this conclusion was reached is not clear from the examples of language transfer provided in the paper.

Author Response

thanks for the very valuable and comprehensive feedback

Reviewer 3 Report

The paper aimed to investigate lecturers’ teaching through English medium of instruction in higher education contexts in China. The topic disused in the paper is timely and is worth investigating. Variety of issues covered by the reported project are also relevant to the field of EMI enquiry. While I enjoyed reading this paper and particularly familiarising myself with its data, I felt that the paper would benefit from further extensive revision prior to being considered for publication in an academic journal. My key points about it are summarised below:  

Abstract is too vague – I suggest that the authors rewrite it, specifying the research problem, design, procedures and outcomes of your specific study. I.e. exclude from the abstract generic sentences, that are also occasionally repetitive.

line.45-46 – please explain what is meant by ‘double first class universities’

line109 – please explain what the difference between CLIL and EMI is

Literature review: There is actually considerable body of classroom-based research that looks into EMI practices in CLIL classrooms. This literature needs to be added into the literature review sections of the paper. Section 2 focuses on the problems caused by EMI teaching but then the section is concluded by critiquing methodological approaches to EMI research. Clearer distinction needs to be made between the how and how each of these themes informs this paper.   

Section 3.2 elaborates extensively on participant observation method, most of this information is unnecessary for an academic publication, while it is relevant for one’s MA or Phd thesis, which this paper is not.

line.189-190 “theoretical tools used are L1/L2 transfer [55], translanguaging [56], pragmatics [26] and dimensional paradigm on teaching and learning between cultures [57].”. Comment – there are some of the KEY concepts that needed to have been discussed in your literature review chapter, in addition to a shorter section on EMI issues.

The research study: further unpacking of the study’s analytical framework (in relation to each of the above) is required in this section.

Findings and discussion – too much data is attempted to be covered in one paper. Leading to fragmented and superficial analysis. I suggest reducing the scope of this paper to a more manageable range of theoretical concepts and analysing them in profound detail, drawing on appropriate theoretical and research literature (that have been introduced and synthesised in the literature review section). As the discussion stands now, it is largely descriptive and draws heavily on the authors’ interpretation of their own data/findings.

4.1. Chinese learners tend to be quiet in the classrooms even when the medium of instruction is Chinese. How representative are then the examples provided in table 1? Is EMI a problem here or something else?  

4.3. what is the difference translanguaging and code-switching?

Conclusion – is too broad. Validity of findings can possibly be questioned due to the fragmented nature of reporting on them in the earlier sections of the paper.

Author Response

greatly appreciate the comprehensive feedback. all the points have been addressed

Round 2

Reviewer 1 Report

The authors have addressed the concern I raised about relevance for this journal; when I did my first review I was not informed by the editorial office that this was part of a special issue.

The authors have not, however, done anything to address my first-round comment about the novelty of the study. Instead, they appear to attempt to discard my comments because I included a link to a Wikipedia article and they consider Wikipedia not a valid source. My purpose in including that link was not to provide an up-to-date scholarly peer-reviewed reference; my purpose was to illustrate that the stuff the authors are claiming as a novel contribution (the fact that language transfer exists) is so well-known that it's been a widely accepted fact for decades and there's a longstanding Wikipedia article about it. No scholarly reference is needed to demonstrate this; a simple Google Scholar search will do. For example, here's an early paper on it:

  • Selinker, L. (1969). Language transfer. General Linguistics, 9.

In any case, the details of which source you look at are irrelevant, and trying to ignore the issue because a reviewer happened to point to a Wikipedia page is disingenuous. The bottom line is that the authors are taking something that has been known for over 50 years and presenting it as a new discovery ("This research found that Chinese background EMI lecturers' classroom English is strongly influenced by their first language."). Thus, it is not clear that this study makes any novel contribution to the field; indeed, it looks to me like the authors never bothered to consult a linguist, who could have told them that this stuff is already common knowledge.

Author Response

Thanks for your time and patience. We have some substantial revision. looking forward to your further advice

Reviewer 2 Report

The key comment for the research presented remains: how are the findings of transfer characterise EMI teaching in particular? As the findings of transfer still seem to characterise any conversation. This concerns the author`s text in line 18 and lines 144-147:

line 18: What features have EMI lecturers demonstrated in the process of their teaching?

lines 144-145:  It will be comprehensive to inspect lecturers’ EMI teaching, beyond language and pedagogy, through dimensional paradigm on teaching and learning between cultures [57]. It seems the idea expressed need to be supported throughout the text in each section of transfer findings.

lines 20-21:  This research reveals that Chinese background lecturers’ EMI teaching is unique in classroom 20 interaction mode, L1. This idea needs development.

lines 137-138: What matters is also appropriate pedagogy. How is this idea supported by the research findings? How is it related to line 148: This research asks: What features have EMI lecturers demonstrated in the process of their teaching?

lines 173-175: Participant observation allows opportunities to observe classroom situations, and to document and analyse the ongoing teaching and learning behaviour of EMI lecturers and their students, including language use, teaching strategies and teacher-students interaction.

What teaching and learning behavior is meant in regard to this research?

line 241:  It is not always related to the lack of confidence in English expression as according to the researchers’ field notes.

The research instrument is not presented and its relation to the linguistic analysis findings related to language transfer performed is not clear.

Section 4.2.1 Grammatical and semantic transfer

Section 4.2.2 Phonological transfer.

It is still not clear how the transfers are characteristic of EMI teaching.

lines 367-368:  The three types of strategies being frequently implemented are signposting, reasoning explanation and repetition.

Thank you for the information added. However, are there any other strategies that characterize EMI teaching as a language phenomenon? And are these the only ones? As these seem to be strategies for any conversation settings.

Perhaps as the authors have put in their cover letter “This study is not quantitative research thus the descriptive data (30%)”, the nature of research and its current stage should be mentioned in the paper in order to avoid further questioning.

Author Response

thanks for your valuable time and patience. we have revised the paper substantially and some the questions from you are not answerable due to the deletion of some old text. we are looking forward to your further advice.

Reviewer 3 Report

Unfortunately, the authors have done only basic revisions to their paper and its merit therefore remains the same as it was at its original submission. My critique points still stand - the paper's focus is too broad, literature review does not not unpack key theoretical areas covered in the paper, discussion and presentation of findings are descriptive. I suggest that in its current form this paper is either rejected or reconsidered after major revision. 

Author Response

thank you for your valuable feedback. it took a little while for us three authors to understand better your constructive feedback. We treated the comments very seriously this time and tried to addressed all the points. We are now looking forward to your further comment!!

Round 3

Reviewer 1 Report

The paper does not identify a clear research question or hypothesis. Note that a TOPIC is not the same thing as a RESEARCH QUESTION or hypothesis. The paper says that it's about EMI in China, but the abstract and the early parts of the introduction never raise a testable yes-no question about that to address. Without that, the paper is not really readable; to be perfectly frank, nobody wants to read pages of history/background without knowing where it is going or what the paper is really about.

  • The lack of clear research question also means it is not clear what GAP in our extant knowledge the paper addresses. In the abstract, the authors claim that "there is little research into EMI in-class teaching and learning" and "lived experiences in EMI in-class practice has been largely". This is clearly false; just a couple minutes on Google was sufficient for me to turn up tons of research on this very thing. (e.g., type in "emi classroom" into Google Scholar and you will see.) I could easily find many papers on EMI that are not just "macro-level" (as the authors characterize the previous literature) but that also examine on-the-ground classroom experiences. e.g., just a few examples:
    Jiang, L., Zhang, L,. & May, S. (2016). Implementing English-medium instruction (EMI) in China: teachers' practices and perceptions, and students' learning motivation and needs. International Journal of Bilingual Education and Bilingualism. 22, 107-119. https://www.tandfonline.com/doi/full/10.1080/13670050.2016.1231166
    Rahmadani, D. (2016). Students' perception of English as a medium of instruction (EMI) in English classroom. Journal on English as a Foreign Language, 6, 131-144. http://e-journal.iain-palangkaraya.ac.id/index.php/jefl/article/view/432
    Hu, G., & Lei, J. (2013). English-medium instruction in Chinese higher education: a case study. Higher Education, 67, 551-567. https://link.springer.com/article/10.1007/s10734-013-9661-5

These are just some of the first few examples I saw; there are many more, it's just not worth my time to painstakingly copy all of them into here (reviewing the literature is the authors' responsibility). My point is that just addressing how the paper is different from these three is not sufficient for publication; there's a large literature already out there, and what I've listed here is just a small sampling of it.
The authors might respond that somewhere in the introduction they do address how their paper is different. But that's not enough. It's not the reader's job to puzzle through the paper and figure out what the authors meant by digging up buried information; it's the author's responsibility to make these points clear and easy to find. I read a couple pages and still couldn't see any indication of what exactly this paper is about or how it is different from any of the tons of previous papers on EMI. Therefore, as a reader, I would not feel like it's worth my trouble to continue reading any further, if the authors themselves can't be bothered to clearly spell out what the point of the paper is.

  • Finally, the argumentation in the paper is rather sloppy, and concepts are defined so vaguely that I don't see how any argument can be put forth about them. I list two examples below; I don't think it would be productive for me to go through and nitpick every sentence, so I stop with these two examples, just to give an idea of the kind of problems that are present throughout the paper.
    The first sentence of the article reads, "Teaching through EMI (English Medium of Instruction) is a fairly new learning delivery system." This is false. English has been a medium of instruction for teaching for pretty much as long as it's existed (e.g., the University of Oxford dates back to about the 12th century, about the same time that Middle English was becoming a thing). I assume the "new" phenomenon the authors are referring to is teaching through EMI *in places where English is not the dominant language*. But the authors don't actually say that. If the paper is this sloppy about what concepts are being discussed, it's not very clear what anyone can learn from the analyses therein.
    "Thus, [EMI] is not a major consideration in countries where English is the national language." -- again, this is false. First of all, the authors seem to be equating "countries or jurisdictions where the first language (L1) of the majority of the population is English" with "countries where English is the national language." These are not the same thing. There are countries where English is the L1 of the majority of the population but English is not the national language (e.g., the United States), and there are many countries where English is a national language but English is not the L1 of most people (see e.g. https://en.wikipedia.org/wiki/List_of_territorial_entities_where_English_is_an_official_language). So once again the authors are mixing up separate things. Furthermore, even in countries where English is the native language of most people, language of instruction is still an important consideration -- e.g., in the United States there have been contentious debates for decades about bilingual vs. EMI vs. native-language schooling for students whose first language is not English.

This is the third time I have reviewed the paper now and I don't see substantial improvement (indeed, the authors haven't even included a response letter detailing the changes they've made; they merely stated "We have some substantial revision."). While the authors seem to de-emphasized the focus on language transfer in accordance with my previous reviews, the paper is still not close to being publishable, for the reasons I have outlined above. It does not seem likely to me that further rounds of revision are going to bring the paper appreciably closer to a publishable state. Therefore, I will not be providing further reviews of this submission.

Author Response

Dear Reviewer,

Thank you for all the arguments throughout the three rounds of review. We would just like to waste a few more minutes of yours to go through our response to a few key points you made: 

  1. testable yes-no question: We do not have explicit testable yes-no question as natural science and psychology research do but we do have a weak version of hypothesis. This hypothesis is reflected in the Multilingual EMI framework. 
  2. The references: you are absolutely right. There are tons of EMI reports in the last two decades since the Bologna Process, and there are some research labelled as studies of classroom practice, but when we got into the details of those papers, we found most of them are about the opinions, perceptions, motivations and beliefs, exactly as those you listed for us. We haven't been able to see what the EMI lecturers' teaching (language use, instruction design etc.) is like. 
  3. False and true issue: we feel powerless and maybe too ambitious to answer true or false questions as academics in humanity, social science and education. The purpose of the research is exploration (improving understanding), description (describing situations and events through observation), and explanation (trying to make sense of how particular phenomenon happened and what might be the reasons behind). We hope this is fine with you and researchers in the field. 

Reviewer 2 Report

The paper seems balanced. Its abstract matches the contents and conclusion.

lines 8o-82 provide the key note to the paper.

lines 8o-82 Literature on EMI teaching tends to be concentrated into three major clusters – a focus on language, a focus on pedagogy or teaching strategies and peripheral studies such as perceptions and attitudes.

The statement given in lines 146-147 can be considered as shallow judgement, please specify what is meant.

lines 146  147  practically there is no research data available on the most effective types of EMI lecturers’ pedagogical preparation programs [29, 27, 70].

Section 5.2.1 Phonological transfer – consonants, vowels and consonant-vowel complex

There is still no certainty provided for the place of phonological transfer in this research. Would authors suggest any conclusions regarding its role in EMI teaching?  Perhaps the authors` response text from point 4 can be expanded.

table 3 minor spell checks required

the section How to make sense of Chinese lecturers’ EMI instruction? has solid referencing but it feels this section needs referencing to the authors findings given in Section 5.2 Pedagogical features

conclusion is meaningful

Author Response

Dear Reviewer,

Thanks again for the further comment. We greatly appreciate the patience and effort you put in to guide us through the journey. We have revised it following your new round of feedback (see the attached).
